# CUS3D: A New Comprehensive Urban-Scale Semantic-Segmentation 3D Benchmark Dataset

Lin Gao, Yu Liu, Xi Chen, Yuxiang Liu *, Shen Yan and Maojun Zhang

College of Systems and Engineering, National University of Defense Technology, Changsha 437100, China;
gaolin0914@nudt.edu.cn (L.G.); jasonyuliu@nudt.edu.cn (Y.L.); xi_chen@nudt.edu.cn (X.C.);
yanshen12@nudt.edu.cn (S.Y.); mjzhang@nudt.edu.cn (M.Z.)
* Correspondence: liuyuxiang17@nudt.edu.cn

**Abstract:** With the continuous advancement of the construction of smart cities, the availability of large-scale and semantically enriched datasets is essential for enhancing the machine's ability to understand urban scenes. Mesh data have a distinct advantage over point cloud data for large-scale scenes, as they can provide inherent geometric topology information and consume less memory space. However, existing publicly available large-scale scene mesh datasets are limited in scale and semantic richness and do not cover a wide range of urban semantic information. The development of 3D semantic segmentation algorithms depends on the availability of datasets. Moreover, existing large-scale 3D datasets lack various types of official annotation data, which hinders the widespread applicability of benchmark applications and may cause label errors during data conversion. To address these issues, we present a comprehensive urban-scale semantic segmentation benchmark dataset. It is suitable for various research pursuits on semantic segmentation methodologies. This dataset contains finely annotated point cloud and mesh data types for 3D, as well as high-resolution original 2D images with detailed 2D semantic annotations. It is constructed from a 3D reconstruction of 10,840 UVA aerial images and spans a vast area of approximately 2.85 square kilometers that covers both urban and rural scenes. The dataset is composed of 152,298,756 3D points and 289,404,088 triangles. Each 3D point, triangular mesh, and the original 2D image in the dataset are carefully labeled with one of the ten semantic categories. Six typical 3D semantic segmentation methods were compared on the CUS3D dataset, with KPConv demonstrating the highest overall performance. The mIoU is 59.72%, OA is 89.42%, and mAcc is 97.88%. Furthermore, the experimental results on the impact of color information on semantic segmentation suggest that incorporating both coordinate and color features can enhance the performance of semantic segmentation. The current limitations of the CUS3D dataset, particularly in class imbalance, will be the primary target for future dataset enhancements.

**Keywords:** urban scene understanding; comprehensive urban-scale dataset; semantic segmentation; benchmark dataset

## 1. Introduction

In the 3D world, the large-scale semantic segmentation of a scene is a task that assigns specific meanings to every object within the 3D scene, aggregating the complex and diverse environmental information into several specific semantic categories to facilitate planning and research. This task helps machines accurately identify object features in large scenes and plays a crucial role in advancing the development of fields such as scene-level robot navigation [1], autonomous driving [2], urban planning [3], spatial analysis [4], and urban fluid simulation [5].

To improve the machine's ability to recognize the semantics behind urban data, researchers often need ample real data from which the machine can learn. Deep-learning-driven algorithms are currently mainstream in large-scale semantic segmentation methods.

With the maturity of convolutional neural networks (CNNs), deep networks that can directly process point clouds (e.g., PointNet [6], PointNet++ [7], KPConv [8], and RandLA-Net [9]) have achieved remarkable results in point-cloud semantic segmentation, especially for city-scale outdoor scenes.

Deep neural networks that directly process mesh data, such as MeshCNN [10] and MeshNet [11], only achieve good results in the semantic segmentation of small-scale objects; they perform poorly in mesh semantic segmentation in large-scale scenes. This phenomenon may be due to the existence of few publicly available large-scale scene mesh datasets with fine-grained semantic annotations. Currently, there is a scarcity of mesh semantic segmentation datasets for outdoor scene , and the ones that do exist have certain limitations. For instance, datasets like ETHZ CVL RueMonge [12] and Hessigheim 3D [13] suffer from multiple labeling errors and unmarked areas. These issues can be attributed to factors like multi-view optimization and indistinct boundaries, leading to label errors in the annotated data. SUM [14] only contains one type of data (mesh) and has only six semantic categories. Thus, it lacks the richness of rural and suburban scene information, making it unsuitable for research on smart urban and rural development planning. Multi-view-based methods for semantic segmentation in large scenes [15] are also among the mainstream approaches for 3D semantic segmentation, but no publicly available large-scale 3D semantic segmentation datasets of scenes with relevant multi-view 2D images currently exist, which somewhat restricts the development of multi-view-based algorithms. Therefore, the current large-scale scene semantic segmentation datasets face the following challenges. First, there is a relatively small number of large-scale scene mesh semantic segmentation datasets, and some of these datasets contain large areas with labeling errors or unlabeled regions. Additionally, the semantic types of scenes are limited and do not include various scenes such as cities, suburbs, and rural areas. Second, these datasets do not provide multiple official data types with precise labels, and errors or omissions in data labeling may occur during the process of the sampling and conversion of 3D data such as point clouds and meshes.

Therefore, we propose a new example of urban-level outdoor semantic segmentation of scenes with rich semantic labels and diverse data types. Our dataset, called CUS3D, includes two types of 3D data: point clouds and meshes, which are annotated with both the real color information and semantic labels. We also provide the original high-resolution aerial images used for 3D reconstruction. Fine-grained semantic annotations are performed on both the 3D data and 2D images, creating a benchmark for current research on 2D and 3D semantic segmentation algorithms. The point-cloud data in this dataset consist of approximately 152,298,756 3D points, while the mesh data consist of 289,404,088 triangles. It covers an area of approximately 2.85 square kilometers and includes 10 semantic categories: building, road, grass, car, high vegetation, playground, water, building site, farmland, and ground. At the same time, we conduct benchmark tests on six classic 3D semantic segmentation deep networks to ensure the suitability of the CUS3D dataset. Compared to existing 3D datasets, CUS3D has the following contributions.

1. CUS3D addresses the problems of the insufficient quantity and high error rate of semantic segmentation datasets for outdoor scene mesh. It provides a new benchmark for large-scale urban scene mesh semantic segmentation. Each triangle face in the dataset has undergone multiple manual cross-checks to ensure the accuracy of the semantic labels.

2. CUS3D is a comprehensive urban-scale outdoor scene semantic segmentation benchmark that also provides official point cloud and high-resolution 2D aerial image data. The semantic labels of 2D images are geometrically consistent with the 3D data, avoiding label errors or omissions in the sampling and data transformation process. It is suitable for testing almost all mainstream 3D semantic segmentation algorithms and can also be used for research in areas such as 3D rendering, 3D reconstruction, and remote sensing image semantic segmentation.

3. CUS3D dataset has richer semantics and covers the semantic information of almost all of the urban scenes. CUS3D also includes the semantic information of suburban and

rural scenes, such as farmland and building sites. This benefit provides new opportunities for large-scale urban applications, such as smart cities and urban planning.

## 2. Related Work

Owing to the diversity of acquisition methods and reconstruction techniques, the data types of 3D data are more varied than those of 2D data. Among them, point clouds, meshes, and voxels are common types of 3D data.

### *2.1. 3D Point Cloud Dataset*

Four categories of existing 3D point cloud datasets exist, based on their scale and scene type: (1) object-level 3D point cloud datasets, (2) indoor scene-level 3D point cloud datasets, (3) outdoor road-level 3D point cloud datasets, and (4) large-scale urban-level 3D point cloud datasets.

#### 2.1.1. Object-Level 3D Point Cloud Datasets

These datasets mainly consist of individual objects of different categories. These include ModelNet [16], ShapeNet [17], ObjectNet3D [18], PartNet [19], ShapePartNet [20], ScanObjectNN [21], and the newly released OmniObject3D [22]. These object-level 3D datasets are commonly used for performance competitions with respect to algorithms such as visual classification and segmentation.

#### 2.1.2. Indoor Scene-Level 3D Point-Cloud Datasets

These types of datasets are usually collected using depth scanners. These datasets include ScanNet [23], SUN RGB-D [24], NYU3D [25], SceneNN [26], and S3DIS [27], released by Stanford University. These datasets are widely used in the early development stage of deep networks for directly processing point clouds. They are suitable for testing 3D semantic segmentation algorithms for small scenes.

#### 2.1.3. Outdoor Road-Level 3D Point-Cloud Datasets

These datasets are usually collected through LiDAR scanners and RGB cameras, including ParisLille-3D [28], SemanticKITTI [29], SemanticPOSS [30], A2D2 [31], Waymo Open Dataset [32], Toronto-3D [33], nuScenes [34], CSPC-Dataset [35], Lyft Dataset [36], Oakland3D [37], Paris-rue-Madame [38], iQmulus [39], and KITTI [40]. Additionally, some datasets simulate road scenes through synthesis to obtain more accurate semantic labels, such as the Synthia Dataset [41]. These datasets are often applied in research on autonomous driving, including target recognition and semantic segmentation.

#### 2.1.4. Large-Scale Urban-Level 3D Point-Cloud Datasets

Large-scale urban-level 3D point-cloud datasets are typically collected in two ways. The first way is through aerial LiDAR scanning, including DALES [42], LASDU [43], DublinCity [44], Semantic3D [45], and ISPRS [46]. The second way is to obtain 2D images through UAV oblique photography and then obtain point-cloud data through 3D reconstruction technology, including Campus3D [47] and SensetUrban [48]. This type of dataset is now more often acquired using the second way. Compared to point-cloud data obtained from aerial LiDAR scanning, this technology offers color information, which can better reflect the structure of urban scenes and is beneficial to smart city planning.

### *2.2. 3D Mesh Dataset*

Mesh is a common representation of 3D data. Currently, relatively few publicly available 3D datasets are based on mesh data. We classify them into two categories: (1) object-level 3D mesh datasets and (2) scene-level 3D mesh datasets.

2.2.1. Object-Level 3D Mesh Datasets

This type of dataset is typically composed of various instances of single objects. In this type of dataset, individual parts of a single object are annotated and segmented, and multiple classes of objects in different poses are categorized. Examples of such datasets include Human Body [49], COSEG [50], SHREC11 [51], MSB [52], and mesh-type 3D data from ModelNet [16]. End-to-end semantic segmentation deep networks for 3D Mesh data are still in their early stage of development. Existing methods such as MeshCNN [10] and MeshNet [11] perform semantic segmentation and classification on mesh data by defining different feature descriptors. The open-source methods validate the reliability of the algorithms using object-level mesh data.

2.2.2. Scene-Level 3D Mesh Datasets

Scene-level 3D mesh datasets are typically obtained through high-quality 3D reconstruction techniques and have geometric topology and high-resolution real scene textures. Currently available indoor 3D mesh datasets include ScanNet [23], Matterport 3D [53], Replica Dataset [54], and 3D-FUTURE [55]. However, relatively few large-scale mesh datasets for outdoor urban-level scenes exist. ETHZ CVL RueMonge [12] is the first benchmark dataset provided in mesh format that is related to urban scenes. However, owing to errors in multi-view optimization and fuzzy boundaries, many incorrect labels exist in this dataset. Hessigheim 3D [13] is a small-scale urban scene semantic segmentation dataset that covers an area of only 0.19 square kilometers. The semantic labels of the mesh data are transferred from the point-cloud data labels, so approximately 40% of the area is unmarked. Some non-manifold vertices also exist, making it difficult to directly apply this dataset. SUM [14] is the latest publicly available outdoor large-scale urban-level 3D mesh dataset known. The dataset covers an area of about 4 square kilometers and includes six semantic categories of urban scenes. However, the SUM [14] dataset only contains mesh data, which are relatively limited in terms of data type and semantic categories. The dataset does not include semantic categories of suburban village scenes for urban and rural planning, so it has certain limitations in practical applications.

In summary, most existing publicly available large-scale 3D datasets consist of a single data type and have limited semantic richness. Especially for large-scale mesh datasets, the number of such datasets is small, and some datasets have partial semantic omissions or marking errors owing to semantic annotation strategies, which affect their use. With the continuous development of smart city planning and other fields, a comprehensive 3D semantic segmentation dataset with multiple data types and rich urban scene semantic information has become meaningful. In the following sections, we introduce a new urban-level outdoor large-scale semantic segmentation comprehensive dataset with diverse data types and rich, accurate semantic annotation. Table 1 compares the Urban3D dataset with other existing 3D urban datasets.

**Table 1.** Comparison of existing 3D urban benchmark datasets.

| Name | Year | Data Type | Spatial Size | Classes | Points/Triangles | RGB | Platforms |
|---|---|---|---|---|---|---|---|
| ISPRS [24] | 2012 | Point cloud | - | 9 | 1.2 M | No | ALS |
| DublinCity [31] | 2019 | Point cloud | $2.0\,\text{km}^2$ | 13 | 260 M | No | ALS |
| DALES [29] | 2020 | Point cloud | $10.0\,\text{km}^2$ | 8 | 505.3 M | No | ALS |
| LASDU [30] | 2020 | Point cloud | $1.02\,\text{km}^2$ | 5 | 3.12 M | No | ALS |
| Campus3D [33] | 2020 | Point cloud | $1.58\,\text{km}^2$ | 24 | 937.1 M | Yes | UAV camera |
| SensetUrban [34] | 2021 | Point cloud | $6.0\,\text{km}^2$ | 13 | 2847.1 M | Yes | UAV camera |
| Oakland3D [23] | 2009 | Point cloud | 1.5 km | 5 | 1.6 M | No | MLS |
| Paris-rue-Madame [25] | 2014 | Point cloud | 0.16 km | 17 | 20 M | No | MLS |
| iQmulus [26] | 2015 | Point cloud | 10.0 km | 8 | 300 M | No | MLS |
| Semantic3D [32] | 2017 | Point cloud | - | 8 | 4000 M | No | TLS |
| Paris-Lille-3D [14] | 2018 | Point cloud | 1.94 km | 9 | 143 M | No | MLS |
| SemanticKITTI [15] | 2019 | Point cloud | 39.2 km | 25 | 4549 M | No | MLS |

**Table 1.** *Cont.*

| Name | Year | Data Type | Spatial Size | Classes | Points/Triangles | RGB | Platforms |
|---|---|---|---|---|---|---|---|
| Toronto-3D [19] | 2020 | Point cloud | 1.0 km | 8 | 78.3 M | Yes | MLS |
| ETHZ RueMonge [44] | 2014 | Mesh | 0.7 km | 9 | 1.8 M | Yes | Auto-mobile camera |
| Hessigheim 3D [45] | 2021 | Point cloud and Mesh | 0.19 km$^2$ | 11 | 125.7 M/36.76 M | Yes | Lidar and camera |
| SUM [46] | 2021 | Mesh | 4 km$^2$ | 6 | 19 M | Yes | UAV camera |
| CUS3D (Ours) | 2023 | Point cloud and Mesh | 2.85 km$^2$ | 10 | 152.3 M/289.4 M | Yes | UAV camera |

## 3. Comprehensive Urban-Scale Semantic Segmentation 3D (CUS3D) Dataset

### 3.1. UAV Captures Aerial Images

In the past, 3D data (such as point clouds) were typically obtained by using LiDAR scanners in cars. With the development of UAV tilt photography and 3D reconstruction technology, UAVs have gained significant advantages in mapping. To obtain high-resolution aerial image sequences, we selected the DJI M300 RTK (https://enterprise.dji.com/cn/matrice-300, accessed on 8 June 2023) (as shown in Figure 1a) and equipped it with the advanced Penta-camera SHARE PSDK 102S V3 (https://shareuav.cn/102PV2, accessed on 12 June 2023) (as shown in Figure 1b) for 2D aerial image acquisition. Figure 1 shows the image acquisition equipment, and Table 2 describes the relevant parameters of the SHARE PSDK 102s camera (https://shareuav.cn/102PV2, accessed on 12 June 2023) (as shown in Figure 1b).

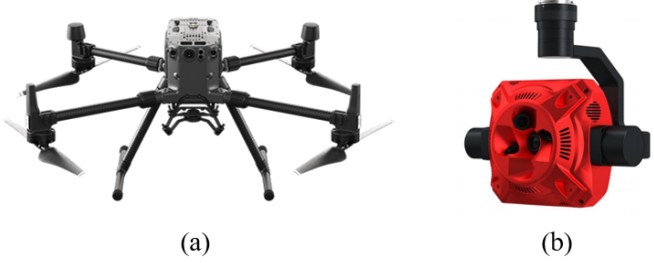

(a)    (b)

**Figure 1.** (**a**) DJI M300 UAV used for data collection; (**b**) SHARE PSDK 102s five-camera used for data collection.

**Table 2.** The parameters of the SHARE PSDK 102s camera.

| Performance Parameter | Numerical Value |
|---|---|
| Lens number | 5 |
| Tilt angle | 45° |
| Image resolution | 6144 × 4096 |
| Focal lens | Downward: 25 mm, sideward: 35 mm |
| Sensor size | APS-C Format (23.5 × 15.6 mm) |

To ensure comprehensive coverage of the designated measurement area, the UAV follows a preplanned Z-shaped flight path and captures images at specified intervals of 2 s. The flight altitude of the UAV is 100 m, and the weather conditions are good without any obstructive factors such as clouds, haze, or strong winds, providing the UAV with good visibility and stable flight conditions. The flight control system is automated. Owing to the limited capacity of the UAV's onboard batteries, every battery group can support approximately 30 min of flight time. The capture of images for the entire area is composed of multiple separate flight missions executed in parallel. Throughout the data-collection process, the UAV acquired a total of 10,840 aerial images from five distinct angles. Ground Sampling Distance (GSD) is a key metric in photogrammetry and remote sensing for

assessing image resolution. In the CUS3D dataset, the original aerial images have a GSD of 2 cm/pixel.

To validate the accuracy and quality of the evaluation data, we use a high-precision, real-time kinematic (RTK) GNSS for geo-referencing the captured aerial images. In addition, six ground control points are manually set up in the real scene to validate the accuracy of the subsequent 3D reconstruction data. The positions of these six checkpoints are the locations where the UAV takes off for data collection. Figure 2 shows the locations of six artificial control points.

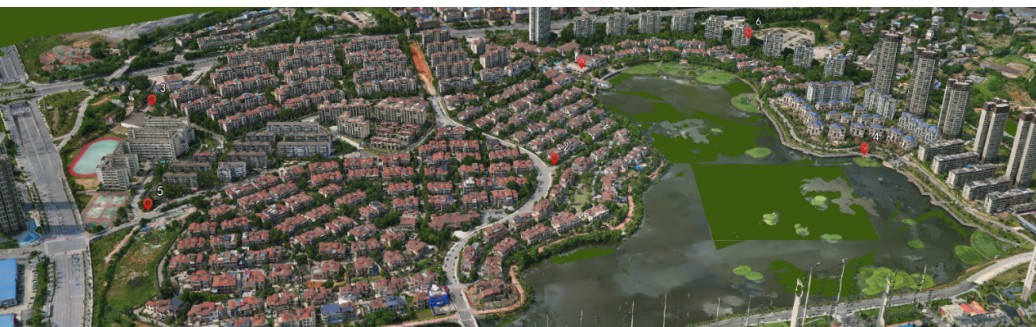

**Figure 2.** When capturing raw aerial images with a UAV, the positions of ground control points are manually set by the personnel on the ground.

*3.2. 3D Model Reconstruction*

In the reconstruction process of mesh data, the first step is to perform aerial triangulation on the collected 2D image data to determine the position and pose of the camera and minimize the re-projection error. Then, the results of the aerial triangulation are used to construct a large-scale 3D scene using 3D reconstruction techniques. The 3D reconstruction process primarily relies on the Structure From Motion (SFM) theory, which is employed to derive 3D information from a sequence of 2D images capturing motion. The process begins with obtaining the mapping relationship between the pixel of the aerial image and the camera imaging plane through camera calibration. Subsequently, the relative pose transformation relationship between the cameras is calculated by performing feature matching between images. The transformation results from multiple image pairs are refined to accurately determine the camera's position and orientation, establishing the corresponding relationship between feature point pixels and 3D points. To identify outliers, the RANSAC algorithm is employed to eliminate matching errors of feature points between images, removing outliers to prevent interference with the pose-estimation process. Finally, the sparse point cloud is projected back onto the image as seed points to generate the depth map, establishing the correspondence between all pixels in the image and 3D points.

The correspondence between 2D pixels and 3D points can be described using affine transformations. The process begins by determining the relative pose transformation between cameras through feature point matching in images. An extrinsic matrix is then created to relate the 3D vertex coordinate system to the camera coordinate system. Subsequently, an intrinsic matrix is constructed using camera parameters like focal length to establish the connection between the camera coordinate system and the pixel coordinate system. Ultimately, the relationship between 2D pixels and 3D points is established through coordinate transformations.

However, large lakes exist in our CUS3D dataset, where the 3D water bodies do not meet the Lambertian assumption during reconstruction. To address this issue, we use semi-automated water body restoration techniques to obtain a complete 3D mesh model. The entire area covers an approximate land area of 2.85 square kilometers and consists of approximately 289 million triangular meshes. Figure 3 shows the dimensions and overall appearance of the reconstructed 3D textured mesh model. The terrain in this area is relatively flat, typical of suburban urban scenes, with a diverse range of features including buildings, roads, water systems, farmland, and vegetation. Note that the entire

reconstructed scene is divided into 93 tiles, but only 89 tiles contain scene data. There are four tiles without any scene data, and their distribution is shown in Figure 4. These four blank tiles will be ignored in subsequent labeling and experiments. The details of the reconstructed mesh and point-cloud scene are shown in Figure 5.

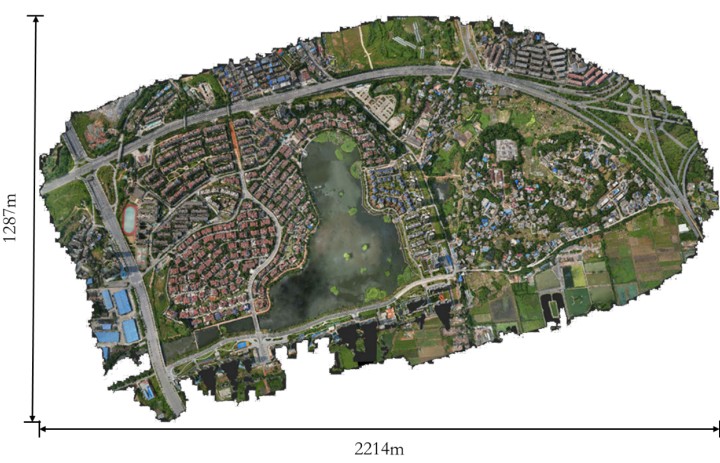

**Figure 3.** Original unmarked 3D texture mesh model. High-resolution texture mesh model reconstructed in 3D from a sequence of aerial images taken by a UAV, covering an area of 2.85 square kilometers.

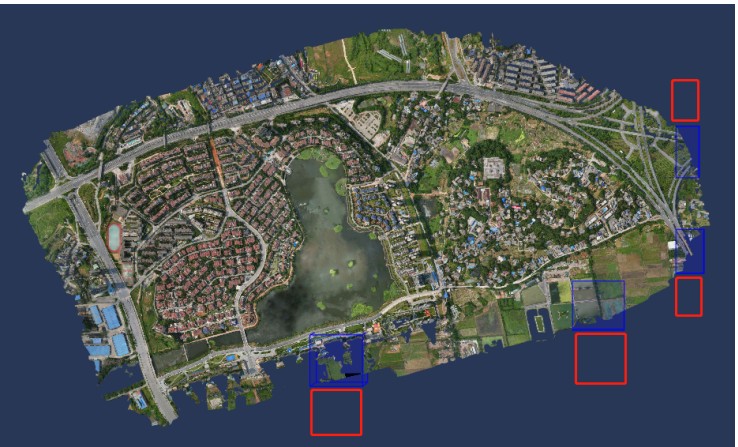

**Figure 4.** Blank Tile Distribution Chart (as indicated by the red box in the image) and Ignored Tile Distribution Chart (as indicated by the blue box in the image).

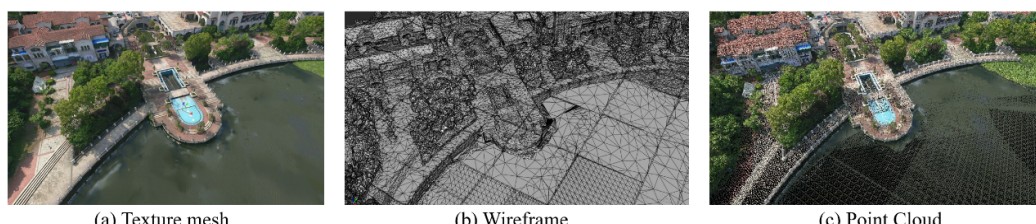

(a) Texture mesh      (b) Wireframe      (c) Point Cloud

**Figure 5.** Three-dimensional reconstruction data display: (**a**) 3D reconstructed textured mesh; (**b**) texture-less mesh with white base model; (**c**) 3D point cloud reconstructed by sampling.

In the reconstruction process of point-cloud data, we down-sample the mesh data to obtain point-cloud data. We transfer the real color information to the vertices of the point-cloud data through the nearest neighbor search. The generated dense colored point cloud consists of 152,298,756 3D points. The point cloud density is approximately 0.15 m, covering an area of about 2.85 square kilometers.

To verify the accuracy of the 3D reconstruction data, we place six control points manually on the ground while capturing the original images. After completing the construction of the 3D model, we report a median re-projection error of 0.82 pixels and a median reprojection error for every control point of 0.62 pixels, which falls within a reasonable error range. Therefore, we consider the constructed 3D model to be accurate and reliable.

### 3.3. 3D Semantic Label Annotation

Our CUS3D dataset contains a variety of data types and diverse semantic information. It consists not only of semantic labels for urban scenes but also semantic labels for rural scenes, such as agricultural landscapes and suburban infrastructure. As modern cities grow and smart city initiatives expand, it has become increasingly important to incorporate urban–rural and suburban planning and construction into smart city development. By integrating the semantic information from rural scenes into urban-level 3D models, we can make accurate, informed decisions for smart urban-rural planning.

In the selection of semantic labeling strategies, the CUS3D dataset abandons the previous fixed urban semantic labeling strategy and adopts a dynamic urban semantic labeling strategy that considers both fully developed and developing semantic categories. For example, the categories of "road" and "ground" may have some overlapping characteristics. However, "road" belongs to fully developed functional objects, which can be used for research on urban transportation planning and peak traffic control optimization. "Ground" belongs to undeveloped objects and labeling, and recognition can help with early planning judgments in urban development. "Building sites" belong to the category of ongoing semantic objects and can be transformed into buildings in future semantic updates. The semantic labeling strategy of the CUS3D dataset considers the application, functionality, and temporal aspects of objects, making it more suitable for practical applications such as urban planning, transportation planning, and construction decision-making.

According to the standard definition of semantic categories, we identify 10 meaningful semantic categories in the CUS3D dataset. Considering that the scarcity of certain objects does not affect the planning of and research on large-scale scenes, we categorize some high-granularity object information (e.g., pedestrians, utility poles, and solar panels) into their respective larger categories. These 10 semantic categories comprehensively represent the scene information in cities and suburbs. Every semantic label is assigned a specific color information. Table 3 provides the RGB values and grayscale values corresponding to every semantic label. Every 3D point or triangle mesh in the dataset is assigned a unique semantic label using annotation tools. The specific 10 object classes in our benchmark dataset are as follows:

1. Building, including commercial buildings and residential homes;
2. Road, including streets, asphalt roads, nonmotorized lanes, and parking lots;
3. Car, including small cars and large buses;
4. Grass, including lawns and small shrubs;
5. High vegetation, including fir trees and banyan trees;
6. Playground, including basketball courts, athletic tracks, and amusement park;
7. Water, including lakes, and canals;
8. Farmland, including cultivated fields, undeveloped land, and livestock enclosures;
9. Building sites, including construction material yards, and construction sites;
10. Ground, including cement surfaces, asphalt surfaces, and bare soil surfaces.

**Table 3.** Semantic label color information table.

| Category | RGB Value | Grayscale Value |
|---|---|---|
| Building | (254, 1, 252) | 105 |
| Road | (0, 255, 255) | 179 |
| Car | (200, 191, 154) | 189 |
| Grass | (91, 200, 31) | 99 |
| High vegetation | (0, 175, 85) | 112 |
| Playground | (130, 30, 30) | 0 |
| Water | (0, 0, 255) | 29 |
| Farmland | (140, 139, 30) | 59 |
| Building sites | (30, 140, 201) | 201 |
| Ground | (154, 0, 255) | 75 |

We use the DP-Modeler software version 2.0 for semi-automated 3D annotation to ensure that every triangle mesh is assigned the corresponding semantic labels. The semantic annotation process starts by inputting the source data, configuring the annotation categories according to requirements, and then performing manual labeling. After labeling has been completed, the classified data are subjected to quality inspection. Abnormal data are reclassified, resulting in labeled output data. The entire annotation process is illustrated in Figure 6, and the tool interface used in the semantic annotation process is shown in Figure 7.

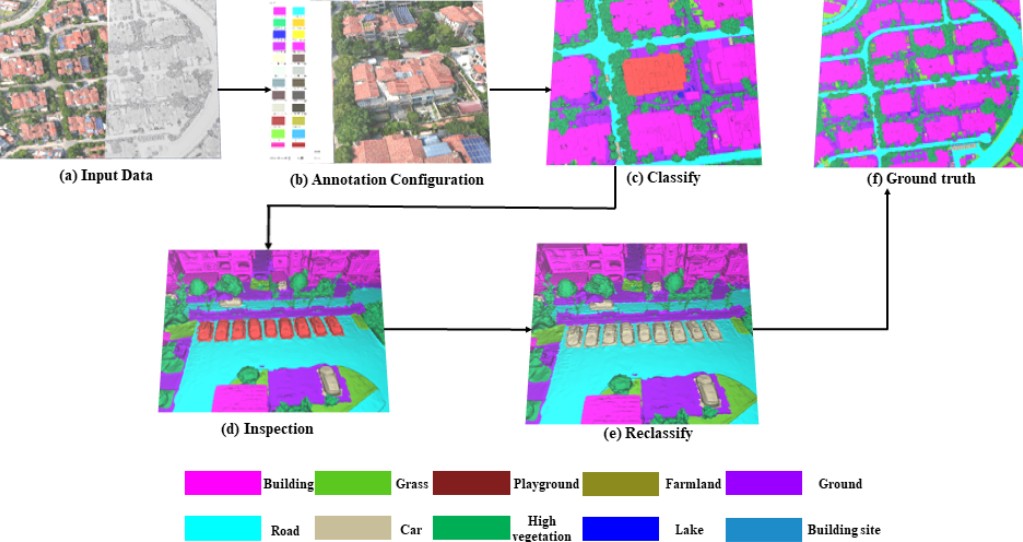

**Figure 6.** The pipeline of semantic tagging work. (**a**) Input the data; (**b**) Configure semantic label categories; (**c**) Assign semantic labels to the data; (**d**) Manually cross-check the labels for correctness; (**e**) Reassign labels for any incorrect ones; (**f**) Complete the data annotation.

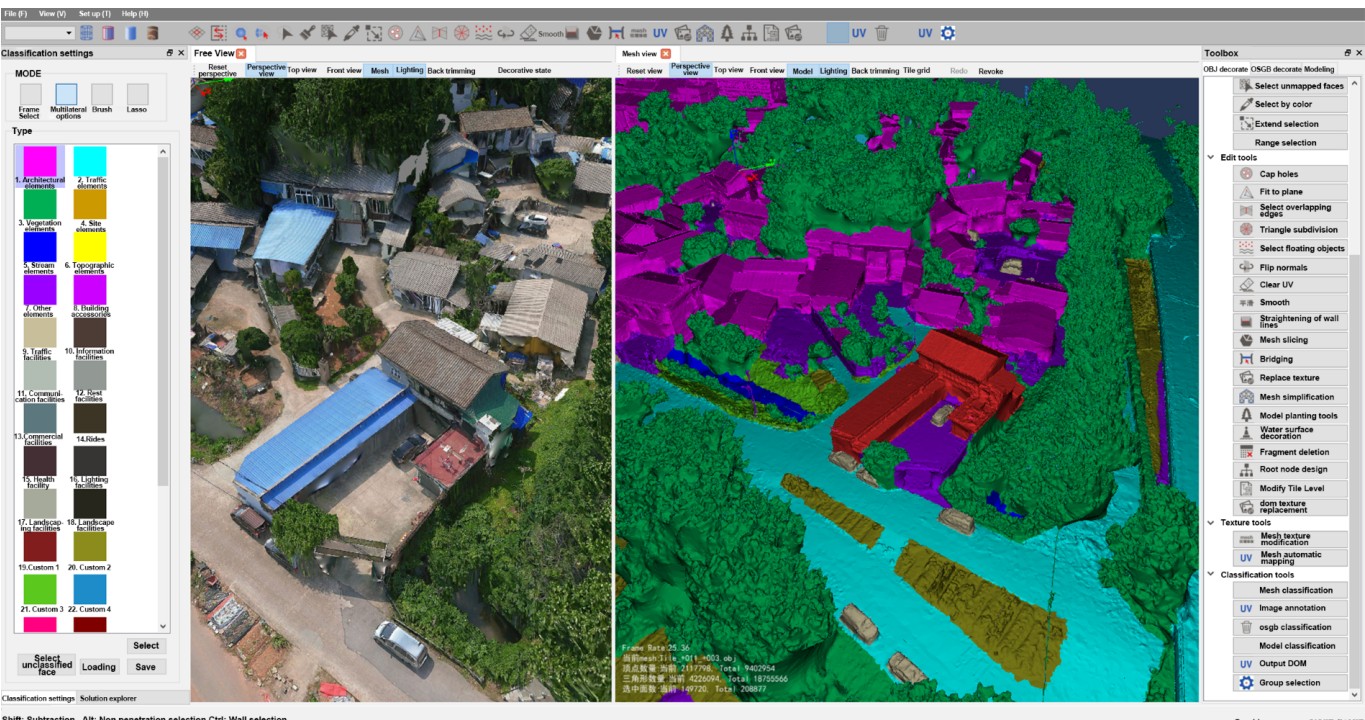

**Figure 7.** Semantic labeling software work interface.

To prevent annotation errors and omissions, we ensure that all of the labels undergo two rounds of manual cross-checking to ensure accuracy. Figures 8 and 9 show the issues that were discovered during the manual cross-checking process and promptly corrected. Figures 10 and 11 show an example of our dataset's semantic annotation.

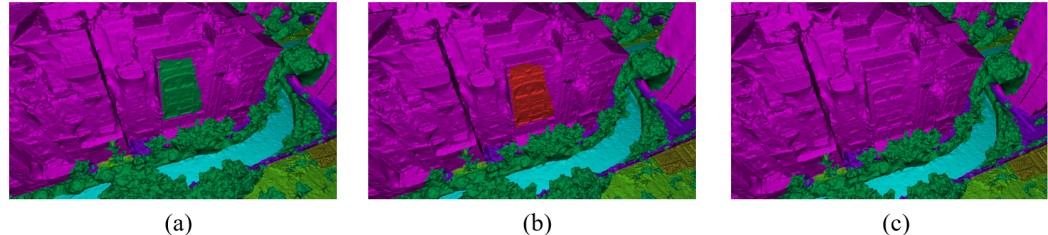

| (a) | (b) | (c) |

**Figure 8.** In the manual cross-check, find and adjust the incorrectly labeled wall. (**a**) Discover the wall that is incorrectly labeled. (**b**) Select the object with the lasso tool. (**c**) Assign the correct label to the incorrectly labeled wall.

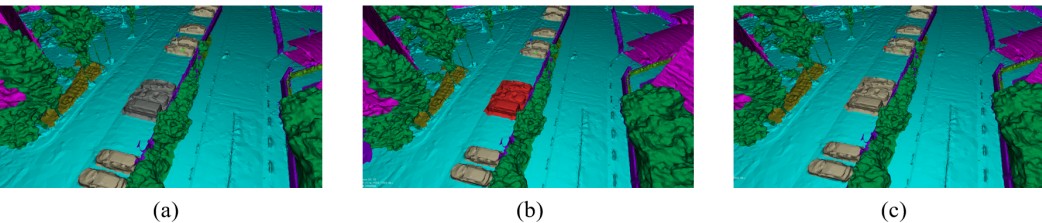

| (a) | (b) | (c) |

**Figure 9.** Artificial cross-checking detection of unmarked vehicles. (**a**) Detect unmarked vehicles; (**b**) select unmarked vehicles; (**c**) change the marking to vehicles.

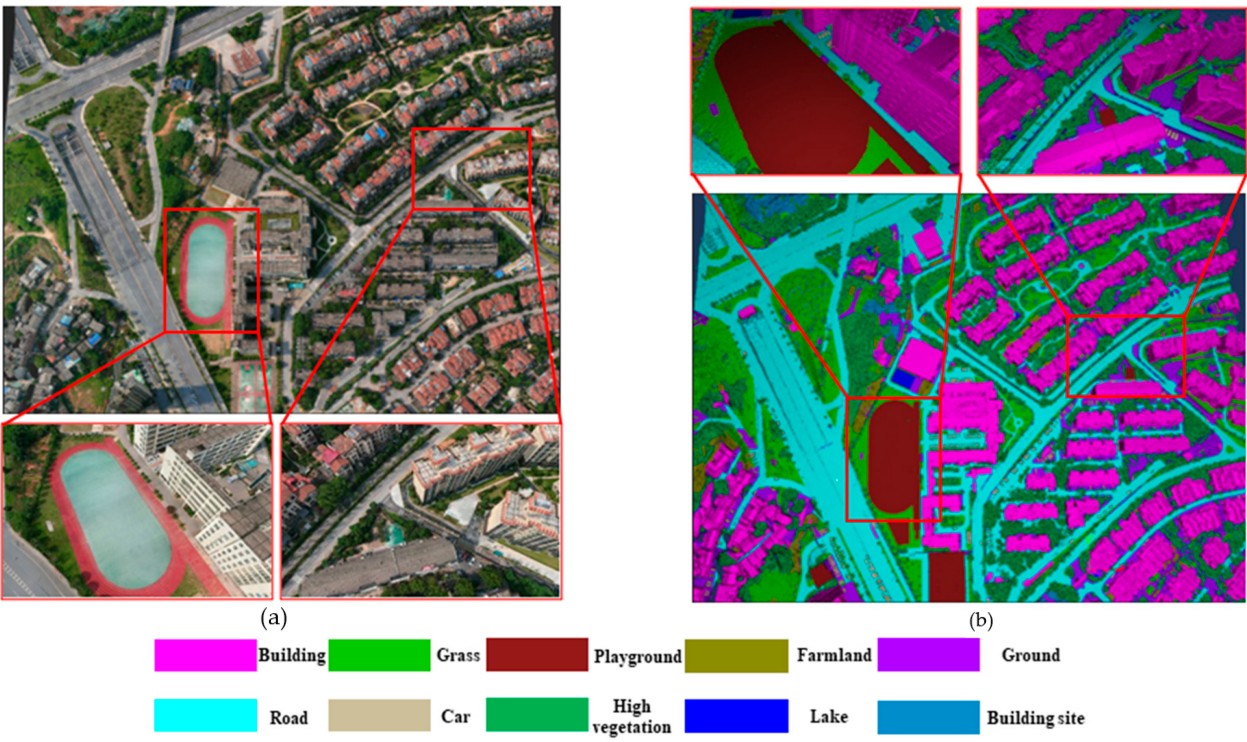

(a)

(b)

| | | | | |
|---|---|---|---|---|
| Building | Grass | Playground | Farmland | Ground |
| Road | Car | High vegetation | Lake | Building site |

**Figure 10.** Partial region semantic labeling results. (**a**) Partial region original mesh. (**b**) Partial region semantic labeling results.

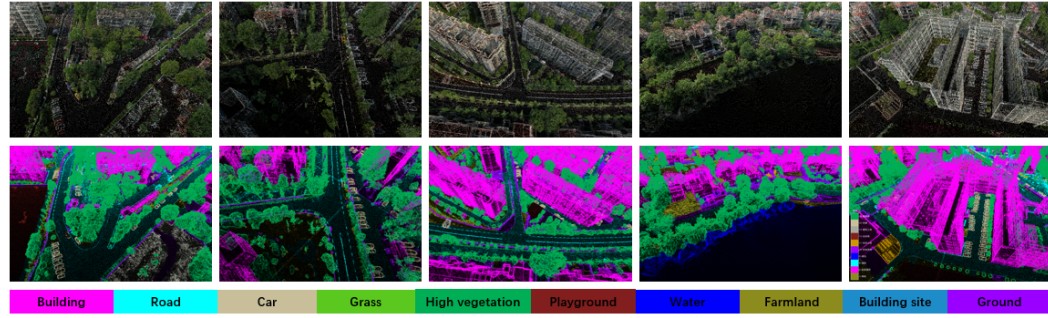

Building | Road | Car | Grass | High vegetation | Playground | Water | Farmland | Building site | Ground

**Figure 11.** CUS3D point-cloud benchmark dataset example.

### 3.4. Two-Dimensional Image Data Annotation

In addition to providing 3D meshes and point-cloud data with rich semantic labels, CUS3D also annotates high-resolution raw aerial images captured by a UAV, giving them detailed semantic labels. There is no publicly available 3D semantic segmentation dataset with annotated 2D images. Providing both 2D and 3D data with semantic consistency is beneficial for the development of 3D rendering, 3D reconstruction, and semantic segmentation of remote sensing images. These finely annotated 2D aerial images are also beneficial for training and improving the recognition capability of classic image semantic segmentation networks like DeepLabV3+ [56] on large-scale aerial imagery.

We use an automated instance segmentation tool based on the SAM [57] model for intelligent semantic labeling. This technology is not limited to specific scenarios and can shorten the labeling cycle. Regarding 2D image semantic labeling, the entire 2D image sequence consists of 10,840 images from five different perspectives. Due to the high similarity in the poses of four cameras tilted at 45°, we only selected 4336 image sequences with a 90° top-down view and one 45° oblique view for semantic annotation. We provide 4336 annotated 2D image data, divided into training, testing, and validation

sets in an 8:1:1 ratio. The semantic labels of these images are the same as the semantic categories of the 3D models. They have geometric alignment in data annotation, and each image is annotated with high-quality pixel-level annotations and manual cross-checking, ensuring high annotation accuracy. Figure 12 shows the results of our 2D aerial image semantic annotation.

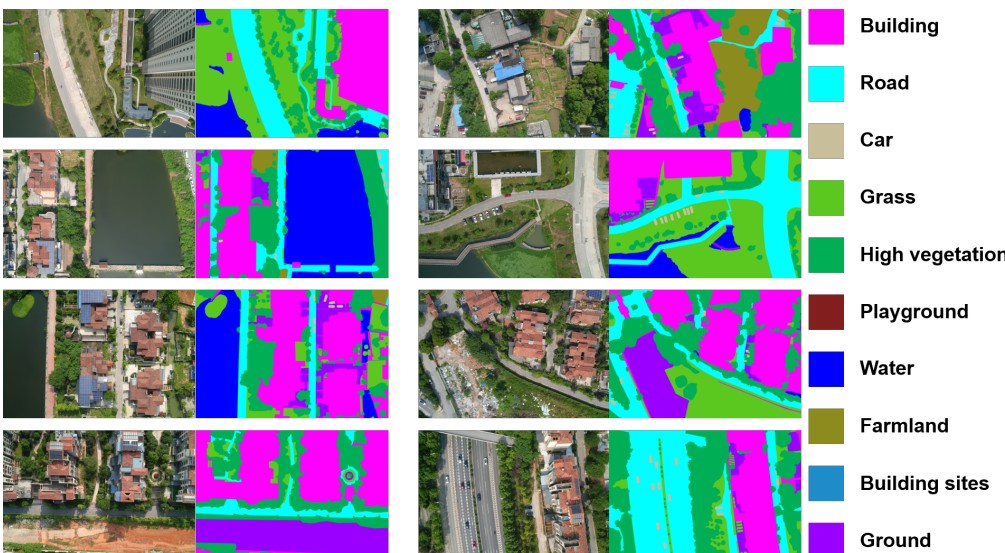

**Figure 12.** Partial high-resolution aerial images and their detailed semantic labels, including 10 semantic categories.

## 4. Experiment

### 4.1. Data Split

During the data partitioning for the experiment, a total of 82 tiles were used for the training/validation/testing sets. Among them, fouur tiles did not have mesh data because they were located at the edge of the measurement area, where the number of feature points for 3D reconstruction was too small to construct mesh data. Additionally, seven tiles were not included in the dataset partition because, during the testing of the RandLA-Net network, the point cloud data of these seven tiles had fewer than 4000 points, which did not meet the network's num_points requirement. In order to ensure the consistency of the input data for the network, these seven tiles were deliberately ignored during the data partitioning.

To use the dataset as a benchmark for semantic segmentation tasks, we divide all of the blocks from the CUS3D dataset in a ratio of 8:1:1. Sixty-six blocks are used as training data, eight blocks as testing data, and eight blocks as validation data (see Figure 13). Owing to the varying sizes of triangles in the 3D mesh data, when evaluating the semantic information of the mesh data, we calculated the surface area occupied by the triangles, rather than the number of triangles. For every one of the ten semantic classes, we calculate the total surface area of the corresponding class in the dataset to represent its distribution, as shown in Figure 13. From the class distribution chart, it can be observed that certain classes, such as the car and playground, account for less than 5% of the total surface area, while categories like building, lake, and high vegetation account for over 70% of the total surface area. Thus, imbalanced class distribution poses a significant challenge in supervised-0learning-based semantic segmentation.

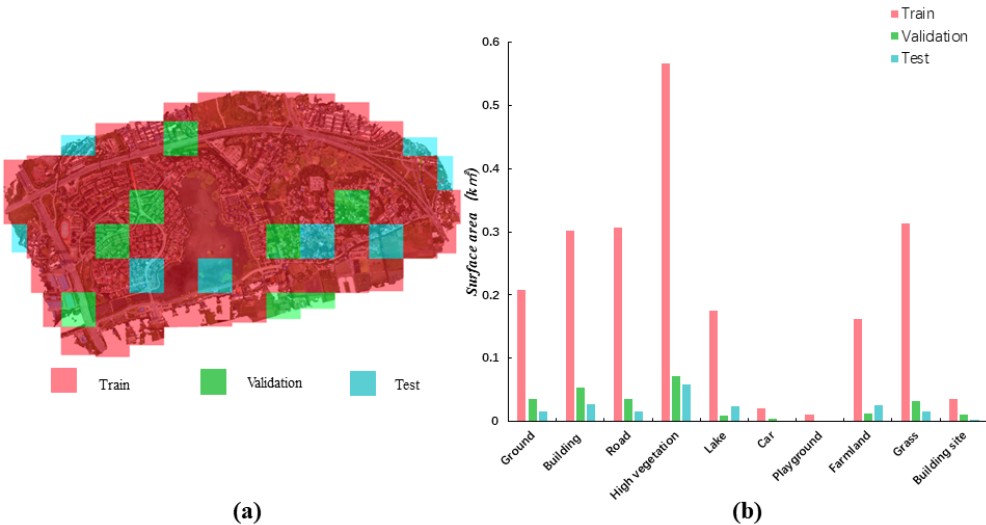

**Figure 13.** Baseline test data partition schematic diagram. (**a**) The distribution of the training, testing, and validation dataset. (**b**) Semantic category distribution in the training, validation, and testing datasets.

### 4.2. Baselines

Currently, to our knowledge, no open-source deep learning framework exists that can directly handle large-scale texture mesh data in urban 3D spaces. However, in urban environments, point clouds and meshes have almost identical fixed attributes. Both types of data share the same set of feature vectors in the same scene. Therefore, we choose point-cloud data from the dataset as the input for testing deep neural networks. The test results show consistent accuracy for both types of 3D data. We carefully select six representative methods as 3D benchmark tests for our CUS3D dataset, including supervised and weakly supervised semantic segmentation methods. Table 4 shows the parameter settings of the baseline experiment. In addition, we selected the DeepLabV3+ [56] network to perform semantic segmentation benchmark testing on 2D aerial imagery data.

1. PointNet [6]. This is a groundbreaking work for directly processing unordered point clouds.
2. PointNet++ [7]. This is an improved version of PointNet that includes the extraction of local features.
3. RandLA-Net [9]. This work directly deals with the semantic segmentation of large-scale point clouds and ranked first in the Semantic3D dataset challenge.
4. KPConv [8]. This method references kernel-point convolution for semantic segmentation and achieves good performance on the DALES dataset.
5. SPGraph [58]. This is one of the methods that uses super point graphs to process large-scale point-cloud data.
6. SQN [59]. This is one of the weakly supervised methods that applies a small number of labels for semantic segmentation of large-scale point clouds.
7. DeepLabV3+ [56]. This is the method for benchmark testing for semantic segmentation of 2D aerial images.

**Table 4.** Baseline experiment parameter settings.

|  | Epoch | Batch_Size | Num_Point | Learning_Rate | Optimizer | Momentum | Parameters | Time |
|---|---|---|---|---|---|---|---|---|
| PointNet | 100 | 24 | 4096 | 0.001 | Adam | 0.9 | 0.97 M | 6.4 h |
| PointNet++ | 200 | 32 | 4096 | 0.001 | Adam | 0.9 | 1.17 M | 6 h |
| RandLA-Net | 100 | 16 | 4096 | 0.01 | Adam | 0.9 | 4.99 M | 7.5 h |
| KPConv | 500 | 10 | - | 0.01 | Adam | 0.9 | 14.08 M | 8 h |
| SPGraph | 500 | 2 | 4096 | 0.01 | Adam | 0.9 | 0.21 M | 5.5 h |
| SQN | 100 | 48 | 4096 | 0.01 | Adam | 0.9 | 3.45 M | 7 h |
| DeepLabV3+ | 200 | 16 | - | 0.007 | Adam | 0.9 | 10.3 M | 8h |

### 4.3. Evaluation Metrics

In the selection of evaluation metrics for semantic segmentation, we chose accuracy, recall, F1 score, and intersection over union (IoU) for every category, similar to the metrics for existing benchmarks. For the overall region testing evaluation, the metrics we adopted are overall accuracy (OA), mean accuracy (mAcc), and mean intersection over union (mIoU). Table 5 shows the baseline test experimental hardware environment configuration.

**Table 5.** Baseline test experimental hardware environment configuration table.

| Name | Model |
|---|---|
| System | 4029GP-TRT2 |
| CPU | Intel Xeon 4210R |
| Memory | SAMSUNG 32 GB DDR4 ECC 293 |
| System Disk | Intel S4510 |
| Data Disk | Intel S4510 |
| GPU | Nvidia Tesla V100 |

### 4.4. Benchmark Results

To ensure fairness in the experiment, we maintained the original experimental environment for every baseline and conducted the experiments on an Nvidia Tesla V100 GPU. The hyper-parameters of every experimental network are adjusted based on the validation data to achieve the best results possible. Table 6 lists the quantitative results of different network experiments after testing. We solely compared the experimental outcomes of the 3D semantic segmentation network. For each class, we chose the best values and emphasized them by highlighting them in bold. The results of the 2D semantic segmentation network are not considered in the comparison. Tables 7–12 show the test results for every semantic category in the six 3D semantic segmentation networks. Figures 14 and 15 show the visual results of some 2D and 3D baseline experiments.

Our CUS3D dataset provides additional color information to the network compared to the large-scale 3D datasets collected by LiDAR, which is helpful for better understanding the real world. However, this may cause the network to overfit. To explore the impact of RGB information on the final semantic segmentation results, we selected five baseline networks for comparative experiments. In every group of baseline comparative experiments, only coordinate information was input, or both the coordinate information and RGB information were input for training. Table 13 shows the quantitative experimental results of different feature inputs on five baselines.

**Table 6.** Benchmark results of the baselines on CUS3D. Overall Accuracy (OA, %), mean class accuracy (mAcc, %), mean IoU (mIoU, %), and per-class IoU (%) are reported.

| | Ground | Building | Road | High Vegetation | Lake | Car | Play-ground | Farm-land | Grass | Building Site | mIou | OA | mAcc |
|---|---|---|---|---|---|---|---|---|---|---|---|---|---|
| PointNet | <u>**78.36**</u> | 65.81 | <u>**78.34**</u> | 73.32 | 25.22 | 20.35 | 8.76 | 8.02 | 39.11 | 16.66 | 42.54 | 74.57 | 94.91 |
| Point-Net++ | 25.78 | 80.36 | 57.77 | 86.18 | 57.52 | 36.13 | 11.38 | 40.88 | 40.17 | 13.28 | 55.16 | 84.52 | 96.90 |
| RandLA-Net | 15.29 | 76.18 | 49.52 | 78.42 | <u>**62.30**</u> | 49.89 | <u>**35.91**</u> | <u>**42.11**</u> | 39.74 | 29.86 | 52.98 | 77.12 | 95.42 |
| KPConv | 25.73 | <u>**82.89**</u> | 48.69 | <u>**89.64**</u> | 48.43 | 44.69 | 25.72 | 41.86 | 44.84 | 15.50 | <u>**59.72**</u> | <u>**89.42**</u> | <u>**97.88**</u> |
| SPGraph | 12.05 | 32.10 | 30.56 | 63.40 | 31.71 | 4.4 | 7.65 | 36.66 | 31.36 | 28.84 | 30.44 | 67.23 | 40.73 |
| SQN | 30.65 | 78.63 | 60.22 | 79.08 | 59.21 | <u>**55.46**</u> | 34.77 | 39.88 | <u>**45.54**</u> | <u>**30.73**</u> | 56.82 | 78.19 | 95.64 |
| DeepLab-V3+ | 81.21 | 80.25 | 82.47 | 92.55 | 65.12 | 59.56 | 38.68 | 46.74 | 45.18 | 40.02 | 63.18 | 90.02 | 98.04 |

The best performance of 3D baseline tests for each semantic category is indicated by bold and underline.

**Table 7.** Evaluation metric results for PointNet for every category.

| | Ground | Building | Road | High Vegetation | Water | Car | Playground | Farmland | Grass | Building Site |
|---|---|---|---|---|---|---|---|---|---|---|
| Acc (%) | 95.96 | 96.47 | 90.44 | 98.03 | 92.14 | 98.77 | 99.90 | 99.90 | 86.72 | 99.18 |
| Recall (%) | 88.35 | 80.95 | 91.77 | 88.80 | 40.42 | 25.32 | 9.98 | 9.98 | 50.53 | 20.10 |
| F1 Score (%) | 87.85 | 79.34 | 87.84 | 84.56 | 40.17 | 33.58 | 16.17 | 16.17 | 56.21 | 28.40 |
| IoU (%) | 78.36 | 65.81 | 78.34 | 73.32 | 25.22 | 20.35 | 8.76 | 8.76 | 39.11 | 16.66 |

**Table 8.** Evaluation metric results for PointNet++ for every category.

| | Ground | Building | Road | High Vegetation | Water | Car | Playground | Farmland | Grass | Building Site |
|---|---|---|---|---|---|---|---|---|---|---|
| Acc (%) | 90.43 | 93.62 | 96.11 | 93.66 | 99.16 | 99.13 | 99.86 | 97.56 | 95.16 | 97.57 |
| Recall (%) | 42.37 | 89.62 | 83.14 | 91.72 | 65.01 | 46.83 | 18.64 | 72.42 | 53.42 | 16.09 |
| F1 Score (%) | 40.77 | 89.07 | 73.07 | 92.55 | 72.45 | 52.61 | 18.67 | 57.50 | 56.89 | 22.58 |
| IoU (%) | 25.78 | 80.36 | 57.77 | 86.18 | 57.52 | 36.13 | 11.38 | 40.88 | 40.17 | 13.28 |

**Table 9.** Evaluation metric results for RandLA-Net for every category.

| | Ground | Building | Road | High Vegetation | Water | Car | Playground | Farmland | Grass | Building Site |
|---|---|---|---|---|---|---|---|---|---|---|
| Acc (%) | 88.24 | 94.49 | 95.12 | 90.15 | 98.67 | 97.64 | 97.32 | 96.67 | 93.31 | 96.59 |
| Recall (%) | 42.56 | 80.52 | 83.02 | 80.44 | 81.33 | 55.31 | 57.31 | 49.56 | 70.68 | 65.00 |
| F1 Score (%) | 26.04 | 86.38 | 65.17 | 87.88 | 76.43 | 66.04 | 51.87 | 58.84 | 56.42 | 45.38 |
| IoU (%) | 15.29 | 76.18 | 49.51 | 78.42 | 62.30 | 49.89 | 35.91 | 42.11 | 39.74 | 9.86 |

**Table 10.** Evaluation metric results for KPConv for every category.

| | Ground | Building | Road | High Vegetation | Water | Car | Playground | Farmland | Grass | Building Site |
|---|---|---|---|---|---|---|---|---|---|---|
| Acc (%) | 93.79 | 95.00 | 97.50 | 93.94 | 99.51 | 99.26 | 99.90 | 98.21 | 96.18 | 98.76 |
| Recall (%) | 35.43 | 92.38 | 80.80 | 95.71 | 60.41 | 58.87 | 13.84 | 62.39 | 54.83 | 17.83 |
| F1 Score (%) | 40.19 | 90.54 | 65.16 | 94.52 | 63.81 | 60.27 | 12.58 | 57.87 | 61.53 | 25.14 |
| IoU (%) | 25.73 | 82.89 | 48.69 | 89.64 | 48.43 | 44.69 | 25.72 | 41.86 | 44.84 | 15.50 |

**Table 11.** Evaluation metric results for SPGraph for every category.

| | Ground | Building | Road | High Vegetation | Water | Car | Playground | Farmland | Grass | Building Site |
|---|---|---|---|---|---|---|---|---|---|---|
| Acc (%) | 92.73 | 69.59 | 96.03 | 21.90 | 99.03 | 98.98 | 99.89 | 97.16 | 92.39 | 98.09 |
| Recall (%) | 14.24 | 38.48 | 42.68 | 90.88 | 40.51 | 4.75 | 8.91 | 51.30 | 42.14 | 42.56 |
| F1 Score (%) | 21.30 | 48.48 | 46.59 | 77.49 | 47.91 | 8.78 | 16.04 | 53.30 | 47.54 | 46.18 |
| IoU (%) | 12.05 | 32.10 | 30.56 | 63.40 | 31.71 | 4.40 | 7.65 | 36.66 | 31.36 | 28.84 |

**Table 12.** Evaluation metric results for SQN for every category.

| | Ground | Building | Road | High Vegetation | Water | Car | Playground | Farmland | Grass | Building Site |
|---|---|---|---|---|---|---|---|---|---|---|
| Acc (%) | 86.09 | 95.36 | 95.06 | 91.98 | 98.28 | 98.04 | 97.87 | 96.52 | 93.22 | 96.45 |
| Recall (%) | 40.26 | 90.12 | 72.07 | 95.92 | 71.00 | 62.74 | 73.31 | 47.47 | 64.61 | 65.87 |
| F1 Score (%) | 46.44 | 87.76 | 74.74 | 87.73 | 73.81 | 70.70 | 50.94 | 56.37 | 61.96 | 46.32 |
| IoU (%) | 30.65 | 78.63 | 60.22 | 79.07 | 59.21 | 55.46 | 34.77 | 39.88 | 45.54 | 30.73 |

**Table 13.** Quantitative results for the five selected baselines on the CUS3D dataset. Overall Accuracy (OA, %), mean class Accuracy (mAcc, %), mean IoU (mIoU, %), and per-class IoU (%) are reported.

| | OA (%) | mAcc (%) | mIou (%) | Ground | Building | Road | High Vegetation | Lake | Car | Play-ground | Farm-land | Grass | Building Site |
|---|---|---|---|---|---|---|---|---|---|---|---|---|---|
| PointNet (w/o RGB) | 54.17 | 90.83 | 20.43 | 38.67 | 23.47 | 59.20 | 22.58 | 12.17 | 0.72 | 0 | 3.15 | 21.85 | 2.18 |
| PointNet (w/ RGB) | 74.57 | 94.91 | 42.54 | **78.36** | 65.81 | **78.34** | 73.32 | 25.22 | 20.35 | 8.76 | 8.02 | 39.11 | 16.66 |
| PointNet++ (w/o RGB) | 70.03 | 94.0 | 29.62 | 11.35 | 50.64 | 33.73 | 65.32 | 15.49 | 15.13 | 0 | 16.78 | 21.29 | 0.14 |
| PointNet++ (w/ RGB) | 84.52 | 96.90 | 55.16 | 25.78 | 80.36 | 57.77 | 86.18 | 57.52 | 36.13 | 11.38 | 40.88 | 40.17 | 13.28 |
| RandLA-Net (w/o RGB) | 70.19 | 94.04 | 40.66 | 10.15 | 73.76 | 30.69 | 70.98 | 44.39 | 42.87 | 8.06 | 36.44 | 24.52 | 18.32 |
| RandLA-Net (w/ RGB) | 77.12 | 95.42 | 52.98 | 15.29 | 76.18 | 49.52 | 78.42 | **62.30** | 49.89 | **35.91** | **42.11** | 39.74 | 29.86 |
| KPConv (w/o RGB) | 85.05 | 96.27 | 44.52 | 17.46 | 77.12 | 37.55 | 84.41 | 25.68 | 34.07 | 12.56 | 24.53 | 30.43 | 0 |
| KPConv (w/ RGB) | **89.42** | **97.88** | **59.72** | 25.73 | **82.89** | 48.69 | **89.64** | 48.43 | 44.69 | 25.72 | 41.86 | 44.84 | 15.50 |
| SQN (w/o RGB) | 69.31 | 93.86 | 41.25 | 24.47 | 70.50 | 48.93 | 67.12 | 44.24 | 40.05 | 3.76 | 28.47 | 26.57 | 15.32 |
| SQN (w/ RGB) | 78.19 | 95.64 | 56.82 | 30.65 | 78.63 | 60.22 | 79.08 | 59.21 | **55.46** | 34.77 | 39.88 | **45.54** | **30.73** |
| SPGraph (w/o RGB) | 67.23 | 39.40 | 30.12 | 12.54 | 30.49 | 28.61 | 63.62 | 30.85 | 4.29 | 7.04 | 37.46 | 30.72 | 29.17 |
| SPGraph (w/ RGB) | 67.23 | 40.73 | 30.44 | 12.05 | 32.10 | 30.56 | 63.40 | 31.71 | 4.4 | 7.65 | 36.66 | 31.36 | 28.84 |

The best performance for each semantic category is indicated by bold and underline.

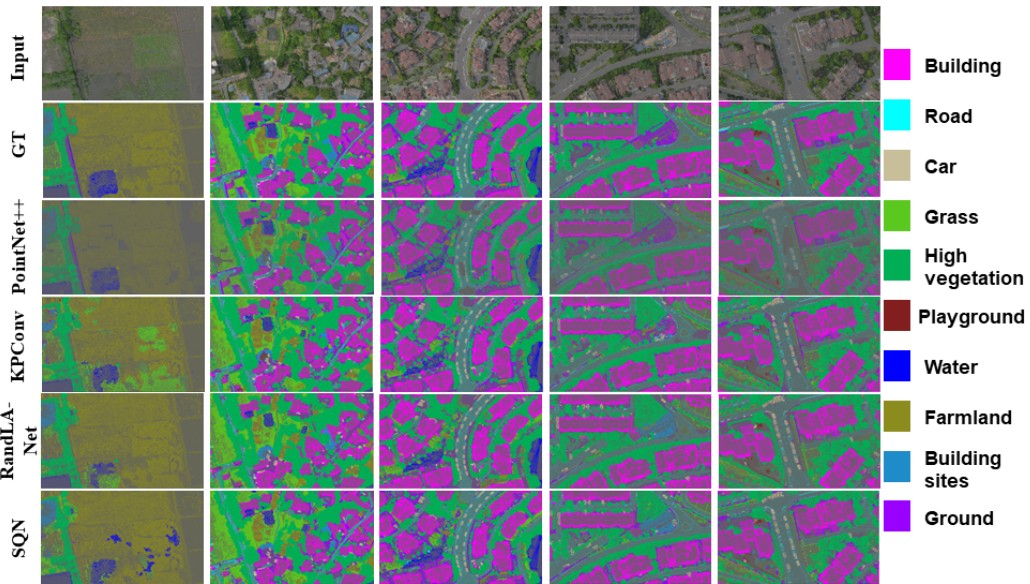

**Figure 14.** Qualitative results of PointNet++ [7], KPConv [8], RandLA-Net [9], and SQN [59] on the test set of CUS3D dataset.

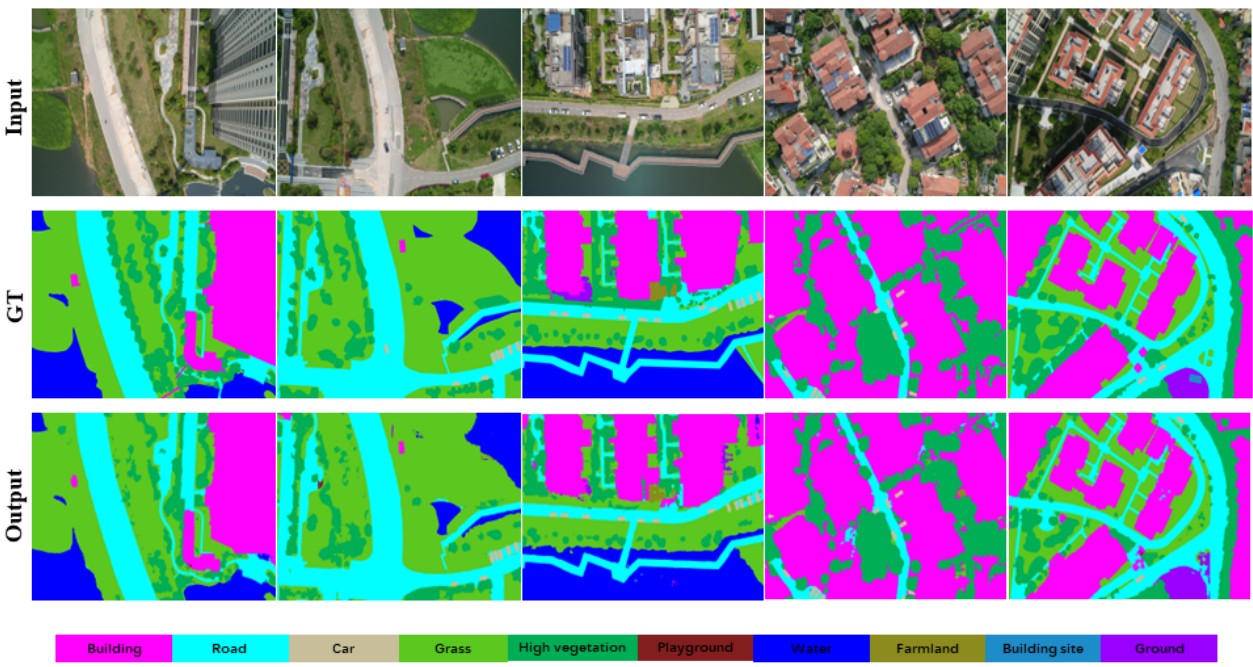

| Building | Road | Car | Grass | High vegetation | Playground | Water | Farmland | Building site | Ground |

**Figure 15.** The results of testing the 2D aerial images in the CUS3D dataset with DeepLabV3+ [56].

## 5. Discussion

On our dataset benchmark, KPConv [8] has the best overall performance in all of the overall performance indicators, with an mIoU of 59.72%, an OA of 89.42%, and a mAcc of 97.88%. The reason KPConv [8] has good overall and individual category performance is that the network has a good ability to learn local features, and it can capture geometric and semantic information in the data. SQN [59] has good overall performance, especially in the few categories with fewer points (e.g., Car, Playground, Building site), thanks to the use of shell-based representations to process point data, which offers good rotation and scaling invariance. At the same time, it can be seen from the results that our dataset does not perform well on SPGraph [58], mainly because it cannot capture local feature information of aggregated data well in large scenes, which also limits the wide application of the CUS3D dataset.

In the experimental results exploring the impact of color information on semantic segmentation, we can see that when coordinate information and color information are input at the same time, PointNet [6], KPConv [8], and RandLA-Net [9] achieve better performance. On the contrary, when only coordinate information is used as input, many urban categories cannot be recognized for segmentation, such as playground, building site, and car categories, which achieve poor segmentation performance. The segmentation performance of SPGraph [58] mainly depends on geometric partitioning, so whether RGB information is available has little effect on its segmentation performance.

The results of the above benchmark test experiment show that our CUS3D dataset can be used as a benchmark for existing 3D semantic segmentation networks, which can help machines better understand 3D scene information. However, the CUS3D dataset currently faces a limitation due to an imbalance in semantic categories within large-scale outdoor scenes. Despite including RGB information, experimental results show significant performance gaps between different categories. For instance, categories such as "Car", "Playground", and "Building Site" may not be recognized in some experiments. Primarily, outdoor scene objects are categorized into buildings, roads, grass, and vegetation, with secondary categories like cars and building sites being less represented. This imbalance in semantic categories could potentially impact the evaluation of semantic segmentation methods in the future. One potential research direction to address this data imbalance could

involve the utilization of more complex loss functions. We firmly believe that researchers will be able to explore an effective solution to address the issue of class imbalance in large-scale scene datasets in the future.

The CUS3D dataset has significantly contributed to the field of semantic segmentation research by addressing the scarcity issue in current outdoor scene mesh semantic segmentation datasets. Through meticulous manual cross-checks, this dataset ensures the accurate assignment of semantic labels to all scene data, expanding the range of semantic categories available for urban and rural scenes. Moreover, the CUS3D dataset offers official point cloud data to prevent inconsistencies in labels due to data conversion in subsequent research. Additionally, it includes original aerial image sequences with annotated semantic labels for their subsequences, making it a valuable resource for research in remote sensing image semantic segmentation, 3D reconstruction, and 3D rendering.

In the future, real-time photogrammetric reconstruction and semantic segmentation are expected to play crucial roles in the intelligent navigation of unmanned aerial vehicles within urban environments. We expect that the CUS3D dataset, as an accurate and high-resolution 3D semantic segmentation dataset, can promote research in intelligent navigation, intelligent transportation, and intelligent cities. Additionally, to enable the application of the CUS3D dataset in more research fields, we plan to supplement it with hyperspectral image data in the future. Hyperspectral images have high spectral resolution and can provide more detailed information about objects, which is beneficial for the inversion of their physical and chemical properties. This will expand the application of the CUS3D dataset to fields such as ocean remote sensing, vegetation remote sensing, precision agriculture, and atmospheric environmental remote sensing.

## 6. Conclusions

This paper has introduced an urban-level outdoor large-scale scene dataset with diverse data types and rich semantic labels. The dataset includes two types of 3D data, point-cloud and mesh, covering an accurately annotated area of 2.85 square kilometers. It consists of 10 semantic labels, encompassing the semantic information of both the urban and rural scenes, serving as a benchmark for 3D semantic segmentation. The dataset provides raw 2D images captured by UAVs, accompanied by detailed semantic labels. These images can be used for research in areas such as 3D reconstruction and high-resolution aerial image semantic segmentation. Through extensive benchmark tests, we have verified the applicability of the CUS3D dataset to different semantic segmentation baselines and conducted experiments and discussions on whether RGB information can improve semantic segmentation performance. In future work, we will continue to expand the scene scale and semantic information of CUS3D. We hope that the CUS3D dataset can promote research progress in emerging cyber-physical fields, such as modern smart city construction and digital twins.

**Author Contributions:** Conceptualization, L.G., Y.L. (Yuxiang Liu). and M.Z.; methodology, L.G. and X.C.; software, L.G.; validation, S.Y.; formal analysis, S.Y.; investigation, L.G.; data curation, X.C.; writing—original draft preparation, L.G. and Y.L. (Yuxiang Liu); writing—review and editing, Y.L. (Yu Liu); visualization, Y.L. (Yuxiang Liu); supervision, Y.L. (Yu Liu); project administration, Y.L. (Yu Liu); funding acquisition, Y.L. (Yu Liu). All authors have read and agreed to the published version of the manuscript.

**Funding:** This research was funded by National Natural Science Foundation of China under Grant 62171451.

**Data Availability Statement:** Our dataset and testing code will be released on https://github.com/CapyLin-G/CUS3D accessed on 31 March 2024 after the paper is accepted.

**Acknowledgments:** This work has been partially supported by the research project of the National Natural Science Foundation of China on cross-category semi-supervised image semantic segmentation methods.

**Conflicts of Interest:** The authors declare no conflict of interest.

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
