# Peer review of "CUS3D: A New Comprehensive Urban-Scale Semantic-Segmentation 3D Benchmark Dataset"

_remotesensing, doi:10.3390/rs16061079_

Round 1
Reviewer 1 Report
Comments and Suggestions for Authors
In this study, a 3D benchmark dataset named CUS3D is presented. If the need in the literature on this subject is taken into account, the study makes a contribution. However, the presented data set has not been sufficiently examined. The dataset was classified with only a few deep learning networks and limited analysis was performed. Here are my other suggestions:
1) The abstract section should be rearranged to provide more information about the manuscript.
2) The original contribution should be explained in the Introduction. There are datasets in the literature that include LiDAR or photogrammetric point clouds. Additionally, data sets have been prepared for urban and rural areas. The importance and contribution of the mesh structure should be stated.
3) What is GSD of the aerial images?
4) In Section 3.2 3D Model Reconstruction: Please mention about Structure From Motion.
5) The images were also labeled but not used in the experiments. An application is also expected to be implemented with labelled images. Otherwise, it is not clear why image labelling was added to the study.
6) The discussion section is not enough. This section should also be developed along with the application.
7) Figures of the classification result should be added. Class-based analysis should be performed and the advantages and disadvantages of the data set should be stated.
Comments on the Quality of English LanguageMinor editing of English language required.
Reviewer 2 Report
Comments and Suggestions for Authors
The authors propose a custom dataset (CUS3D) of UAV imagery specifically designed for 3D semantic segmentation classification of large-scale texture mesh data. The authors detail the current lack of available benchmark datasets for this task, the acquisition of the imagery, annotation and labeling details as well as applying a series of semantic segmentation methods to demonstrate the utility of the CUS3D dataset as an effective benchmark for evaluating the performance of various methods.
The paper is well very written, and I very much commend the authors on their work. Development of remote sensing benchmark datasets like this is much needed, as the remote sensing/earth observation field in general is quite short of datasets using latest semantic segmentation, ML, and DL methods. In addition, the authors releasing their code implementation as well as the imagery in a github, is very much appreciated, and also holds up to the paper’s promise of a freely available dataset. This is also important for purposes of reproducibility as well.
My only major concern would be if this paper qualifies as a full article, as while the dataset itself is novel, the paper is not investigating any novel methods, or providing an innovative evaluation of existing methods. Would this paper be more suitable as a technical report or communication? However, I believe that question should be left to the editors rather than with the reviewers. Otherwise, I think the paper is quite suitable for publication.
Minor Comments:
Section 3.1 – Could you provide some temporal and phenological information about when the dataset was acquired. This might be useful for individuals seeking to use the benchmark dataset for environmental studies.
Ln 302 – Please describe the rationale for choosing a 8:1:1 ratio for training, validation, and test data. The validation and test data are quite small proportions individually. Were the training, validation, and test data blocks randomly chosen?
Figure 13 - The axis on the graph in Figure 13 are very difficult to read. It’s quite difficult to determine what this graph is describing.
The paper could benefit from some light proofreading.
Reviewer 3 Report
Comments and Suggestions for Authors
The authors present an excellent dataset with rigorous tests of its validity on existing segmentation algorithms. I think the manuscript is almost acceptable for publication in Remote Sensing.
Minor corrections:
1. What does CUS3D stand for?
2. Line 51: These should be "these"
Round 2
Reviewer 1 Report
Comments and Suggestions for Authors
I thank the authors for the revisions. The manuscript seems ready for acceptance in its current form.